# *Herbarium Apenninicum* (APP): An Archive of Vascular Plants from Central Italy

Fabio Conti ⬤, Giacomo Cangelmi *⬤, Jamila Da Valle ⬤ and Fabrizio Bartolucci ⬤

Scuola di Bioscienze e Medicina Veterinaria, Università di Camerino–Centro Ricerche Floristiche dell'Appennino, Parco Nazionale del Gran Sasso e Monti della Laga, San Colombo, Barisciano, IT-67021 L'Aquila, Italy; fabio.conti@unicam.it (F.C.); jamiladavalle@gmail.com (J.D.V.); fabrizio.bartolucci@unicam.it (F.B.)
* Correspondence: giacomo.cangelmi@gmail.com

**Abstract:** The *Herbarium Apenninicum* (international code: APP), hosted in the Floristic Research Center of the Apennines (Abruzzo, central Italy), is approximately composed of about 80,000 specimens of vascular plants; 66,352 of them are mounted with data labels and entered in a database. The specimens from the Abruzzo administrative region (central Italy) correspond to more than half of the collection (57.8% of the specimens), while immediately afterwards, other neighboring provinces of central Italy follow. Outside of Italy, the most represented areas are Morocco and southern European countries. Most of the specimens were collected between 2001 and 2020; nevertheless, the herbarium also contains two historical collections from the end of the nineteenth century to the beginning of the twentieth century. The herbarium houses 146 types.

**Keywords:** botanical collection; herbaria; Italy; mediterranean flora; taxonomy; type specimens

## 1. Introduction

The herbaria are the primary means of documenting plant diversity on earth, and they currently assume a fundamental role in the study of genomics and climate change outcomes, detected by monitoring plant distribution over time [1,2]. They have always constituted a key information basis for other more traditional and consolidated fields of study, such as systematics and taxonomy, floristics and phytogeography, ecology, ethnobotany, the history of botany and biodiversity conservation, beyond their educational and cultural meaning, e.g., [3–6]. Currently, around 396 million specimens are kept in 3567 active herbaria [1]. Furthermore, a discussion has recently opened on the concrete possibility of resurrecting extinct-in-the-wild plants (de-extinction) from herbaria [7,8].

The *Herbarium Apenninicum* was established in 2002 by an agreement between the University of Camerino and the National Park of Gran Sasso and Monti della Laga. It has been recognized since 2002 by the Index Herbariorum ([9] see https://sweetgum.nybg.org/science/ih/herbarium-details/?irn=125658, accessed on 10 June 2023) with the international code APP. It is hosted in the Floristic Research Center of the Apennines—Centro Ricerche Floristiche dell'Appennino (hereafter, CRFA). The CRFA is located in the former monastery of San Colombo, inside the National Park of Gran Sasso and Monti della Laga in the municipality of Barisciano (L'Aquila), in the Abruzzo administrative region (Italy) (Figure 1).

Since its creation, the management of the CRFA is entrusted to the University of Camerino. The CRFA was created with the aim of increasing Apennine floristic knowledge; it consists of a structure of excellence for floristic, systematic and taxonomic research in Italy, where it represents a key node in the network of national experts and amateur botanists and is the headquarter of the database of the checklist of Italian vascular flora, producing several updates every year [10]. The *Herbarium Apenninicum* can be considered, due to its consistency and increased level of usability, the most important herbarium in Abruzzo. Among the most recent results from the floristic point of view are the checklists and the

photographic guides of the vascular flora for three national parks in central Italy; the National Park of Abruzzo, Lazio and Molise; the National Park of the Maiella; and the National Park of Gran Sasso and Monti della Laga [11–13]. From a taxonomic point of view, 29 new taxa were described, and around 100 new names and combinations were proposed.

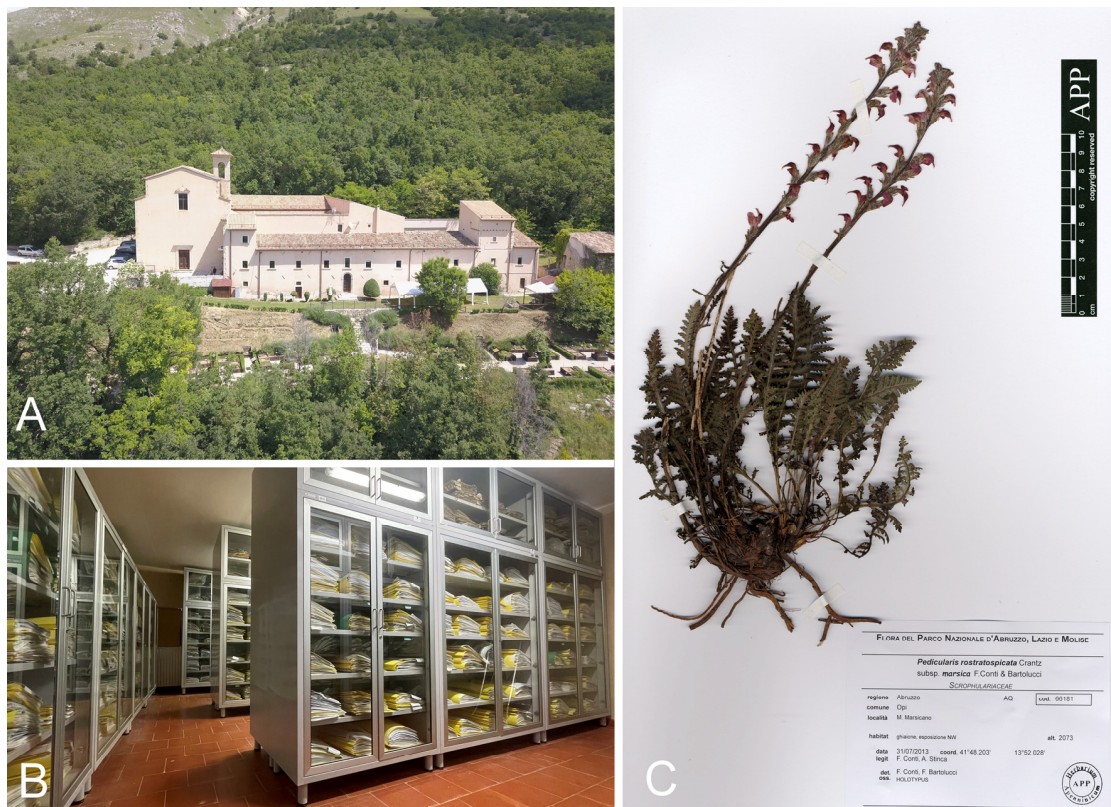

**Figure 1.** (**A**) Monastery of San Colombo (Barisciano, L'Aquila; photo by R. Marchesan); (**B**) room of APP (photo by F. Conti); (**C**) holotype (APP n. 66181) of *Pedicularis rostratospicata* Crantz subsp. *marsica* F.Conti & Bartolucci.

We estimate that the APP herbarium is approximately composed of about 80,000 specimens, of which 66,352 (82.94%) are mounted with data labels and entered in a database that includes collection data and very few images of scanned specimens, which are, at the moment, not accessible online. The activity of the CRFA leads to continuous field research and new specimen collections every year, and so the numbers presented here are to be considered as subject to constant updates.

## 2. Materials and Methods

Plant samples are collected in field investigations conducted with different purposes, including floristic inventories, systematic and taxonomic revisions and conservation projects. Sample preparation is carried out by drying and pressing the plants placed inside a newspaper sheet, with each of them between absorbent paper alternated with perforated cardboard. This structure is then placed under a warm air flow produced by a fan heater for around 4 h in order to remove all moisture. The dried samples are determined to the finest taxonomic level possible with the aid of national and international specialized literature; after that, specimens are pasted to the herbarium sheet and accompanied by the herbarium labels, reporting all the relevant information about the sample: geographical information (coordinates, municipality, province, administrative region, country), habitat and altitude of the site of collection, date of collection and collector names. These data are entered into a database prepared with the FileMaker Pro 19.2.2.234 software. Concerning central Italy, a specific project GIS for floristic cartography has also been studied and implemented.

To avoid attacks by bacteria, fungi or insects, the prepared specimens are stored in a controlled environment at a constant temperature of 16 °C and below 35% air humidity. To prevent the development of damaging organisms, the major parts of the specimens are always located inside shelves with protective sheathing. Conservation through low temperatures is facilitated by the location of the monastery at around 1100 m above sea level. Before being inserted into the two rooms that house the herbarium, the samples are frozen (−30 °C) for 48 h to facilitate the conservation of the material, thus avoiding insect or fungal attacks. This process is also repeated continuously on the samples that are already inserted [14].

The specimens are organized into four groups and placed in two rooms (ferns and fern allies, gymnosperms, angiosperms—monocots in one and angiosperms—dicots in the other). Within each group, the specimens are ordered alphabetically in the following sequence: families, genera, species, subspecies, varieties and forms. Each specimen has a unique accession number.

The nomenclature follows the checklists of the Italian native [10] and alien [15] vascular flora and is continuously updated according to the Portal to the Flora of Italy (PFI) ([16]; see also [17]). For species not present in Italy, the nomenclature follows Plants of the World Online (POWO) [18]. Finally, the nomenclature of certain specimens follows recent publications not yet implemented in the PFI or the POWO.

## 3. Results

### 3.1. Taxonomic Coverage

Among the 66,352 digitized specimens, almost all are identified at the species or infraspecific level, 9% at genus level, 0.66% are attributed to groups and 0.44% represent hybrid entities. The APP herbarium hosts a total number of 4562 species and subspecies. The specimens preserved in the APP herbarium belong to the class Magnoliidae for the great majority (97.19% of the specimens), as well as to Polypodiidae (1.85%), Piniidae (0.45%), Equisetiidae (0.43%) and in lower part to Lycopodiidae (0.08%). The specimens present in the APP herbarium represent, to date, 164 families, with the prevalence of Asteraceae Bercht. & J.Presl (15.8%), followed by Fabaceae Lindl. (8.7%), Poaceae Barnhart (7.5%), Caryophyllaceae Juss. (6.1%), Lamiaceae Martinov (4.8%), Brassicaceae Burnett (4.2%), Rosaceae Juss. (4%), Apiaceae Lindl. (3.5%), Ranunculaceae Juss. (3.4%), Plantaginaceae Juss. (2.8%), Cyperaceae Juss. (2.7%) and Rubiaceae Juss. (2.4%) (Figure 2).

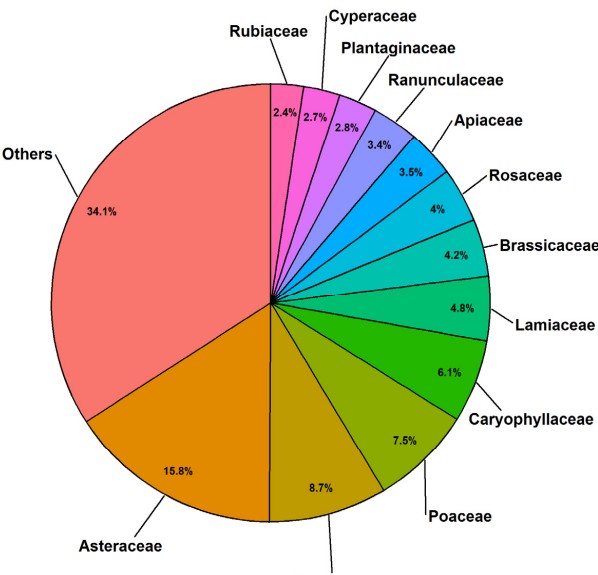

**Figure 2.** Taxonomic coverage of the most represented families in APP. Families with less than 2% are grouped in "others".

The herbarium includes 1111 genera, of which the more significant ones are *Centaurea* L. (2972 specimens), *Carex* L. (1478 specimens), *Ranunculus* L. (1206 specimens), *Galium* L. (1071 specimens) and *Silene* L. (1022 specimens) (Figure 3).

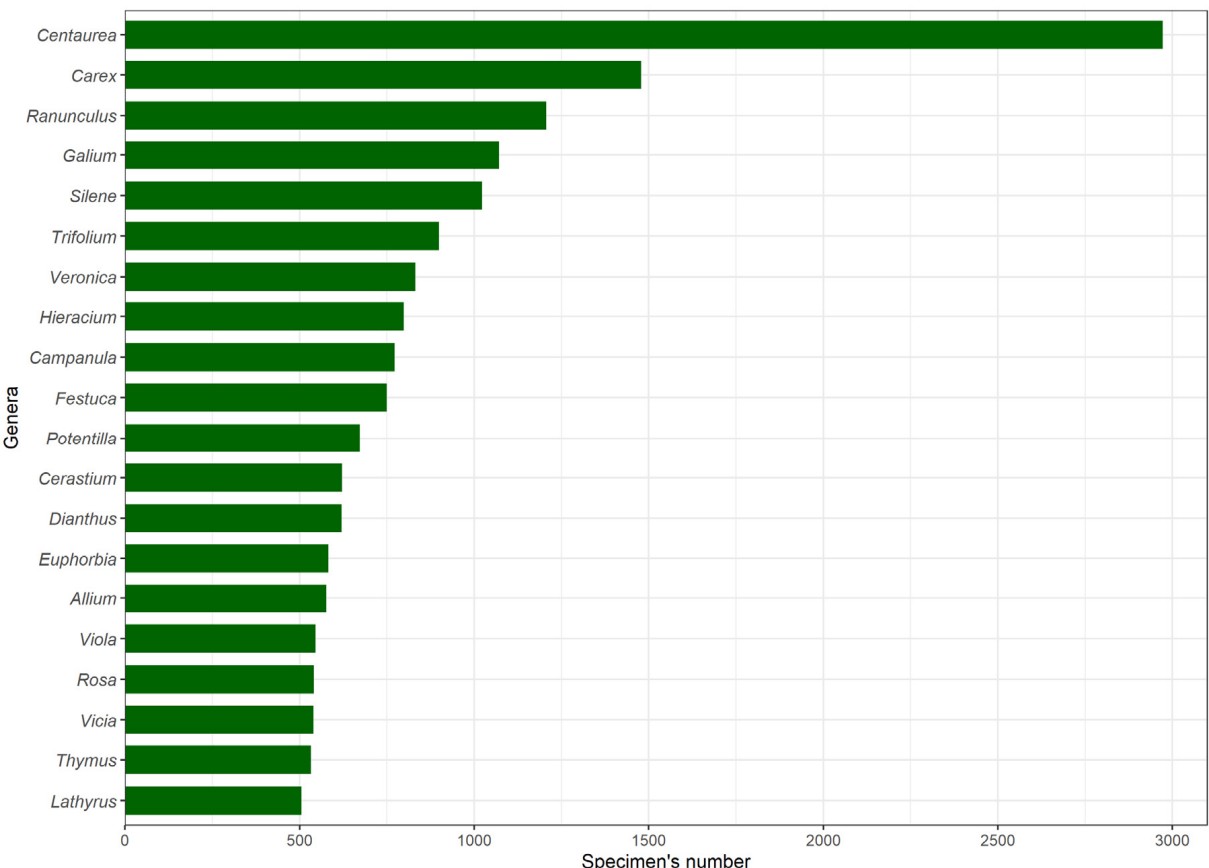

**Figure 3.** Taxonomic coverage of the most represented genera in APP (number of specimens higher than 500).

*3.2. Geographical Coverage*

The specimens housed in the APP were collected from different countries, mainly belonging to the Mediterranean Basin, with a specific focus on Italy, in central Apennines and the Balkan Peninsula (a considerable number from Bosnia–Herzegovina and Greece are not yet digitized). The geographic distribution of the specimens reflects the great taxonomic, floristic and phytogeographical interest for plant species with an amphi-Adriatic chorotype. As depicted in Figure 4, the specimens from the Abruzzo administrative region correspond to more than half of the collection (57.8% of the specimens), while other regions from central Italy and most of the central portions of the Apennines chain follow on immediately: Molise (7%), Lazio (4.1%) and Marche (3.7%). Outside of Italy, the most explored countries are Morocco (1335 specimens), Croatia (1133 specimens), Spain (1043 specimens), Montenegro (862 specimens), Albania (784 specimens), Greece (749 specimens), Bosnia–Herzegovina (626 specimens) and Turkey (537 specimens); Portugal, France, Slovenia, the Czech Republic, Romania, San Marino, Bulgaria, Slovakia, Estonia, Serbia, Switzerland, Lebanon, North Macedonia, Germany and Austria are also represented, although with a number of specimens lower than 500 (Figure 5). Despite the prevalence of administrative regions from central Italy, when observed at a finer spatial scale, the distribution of specimens highlights some other areas of interest, like the 1277 specimens from the province of Potenza (Basilicata) or the 830 specimens from the province of Trento (Trentino—Alto Adige) that exceed the number of specimens from Campobasso (Molise), Latina, Viterbo (Lazio) and Macerata (Marche).

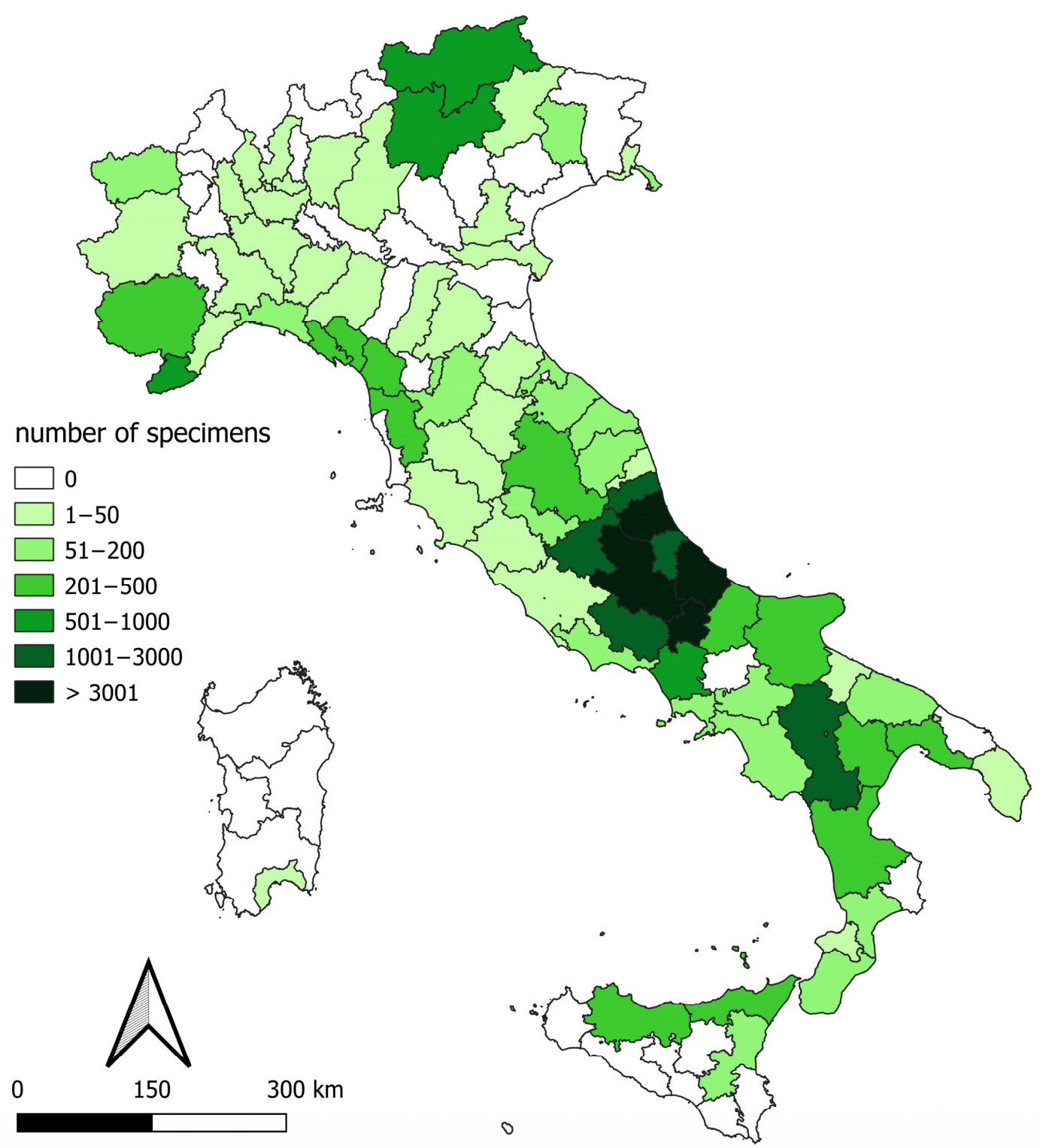

**Figure 4.** Italian geographic coverage of the specimens preserved in APP. The number of specimens is calculated for the Italian administrative provinces.

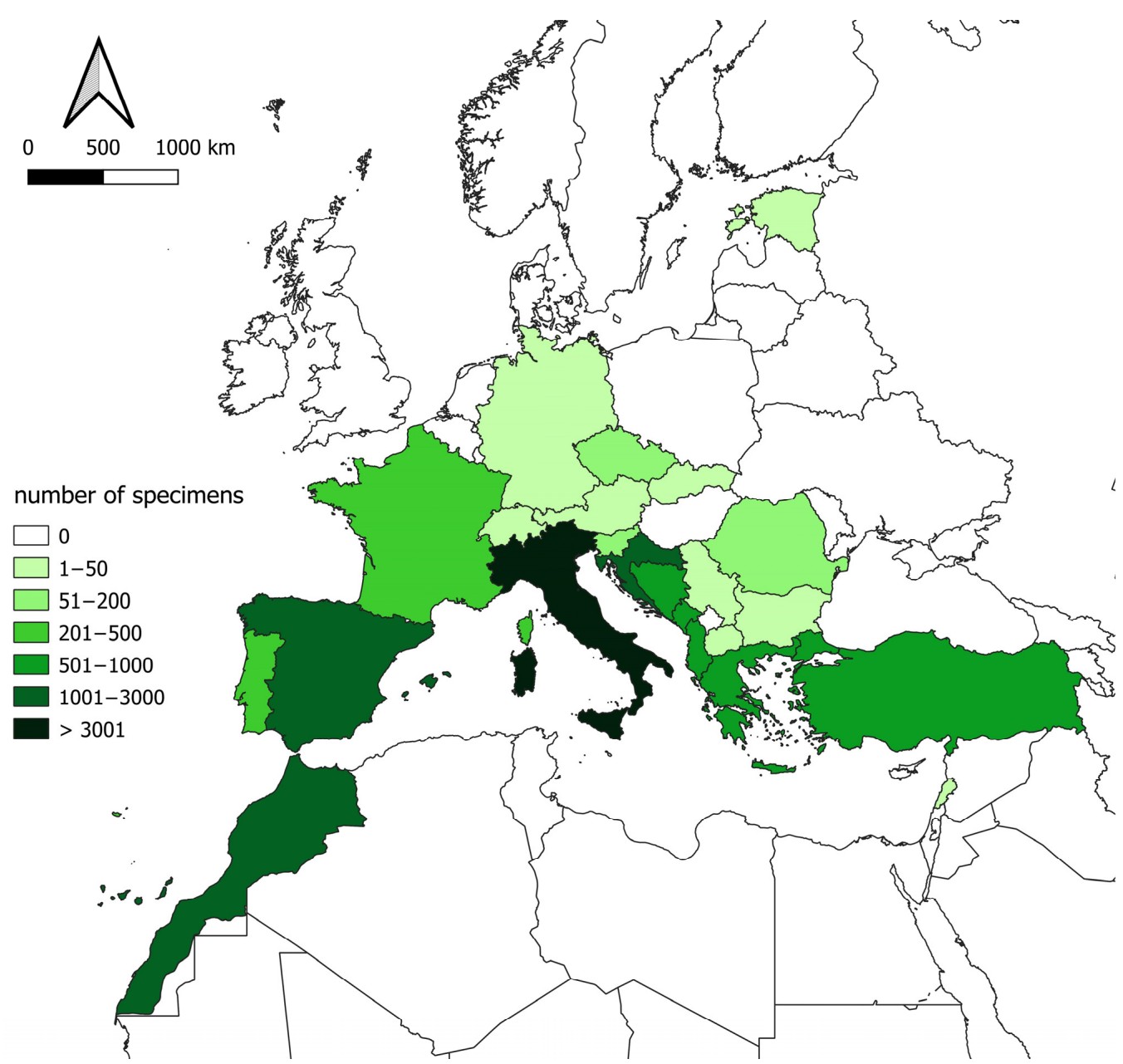

**Figure 5.** Geographic coverage of the specimens preserved in APP.

### 3.3. Temporal Coverage

The APP herbarium is also noticeable for the presence of several historical specimens, among which "Herbarium Zodda" (1894–1965) and "Herbarium D'Amato" (1867–1890) are the most conspicuous, belonging to known past botanists, respectively, with 416 and 112 specimens. These historical collections were found at the institute "V. Comi" of Teramo and donated to the *Herbarium Apenninicum* [19]. The APP also hosts other ancient collections which lack certain information: 660 samples previously owned by the high school "M. Delfico" of Teramo, dating back to the period 1872–1929, and 34 specimens belonging to "Herbarium Quartapelle", probably referable to the second half of 19th century [20]. This last collection includes three specimens of taxa extinct in Abruzzo [*Schoenoplectus triqueter* (L.) Palla, *Achillea maritima* (L.) Ehrend. & Y.P. Guo subsp. *maritima* and *Dracunculus vulgaris* Schott.]. The year of specimen collection of the entire herbarium ranges from 1867 to the present day. The period of major activity of the CRFA lies between 2001 and 2010, with a mean of more than 3000 samples collected every year. After this period, the number of

collections decreased due to the abundant knowledge previously acquired and the more targeted field investigations as a consequence, with a mean of 1030 specimens per year collected between 2011 and 2020 (Figure 6).

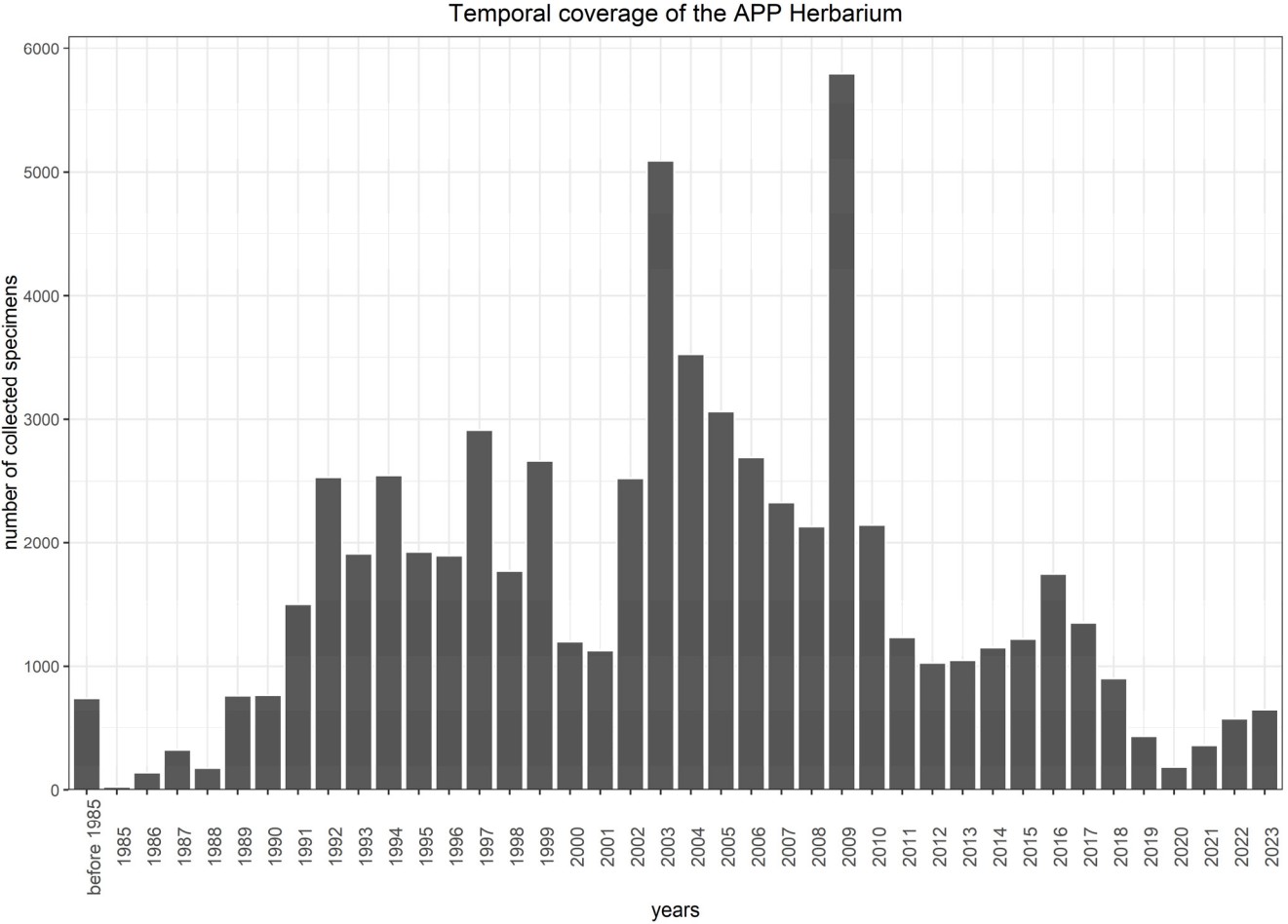

**Figure 6.** Bar plot showing the number of specimens preserved in APP by year of collection; samples collected before 1985 are grouped in the first bar.

### 3.4. Interest of Collections

According to Taffetani [3], the APP herbarium was, in 2012, the 27th herbarium in Italy by number of preserved samples, with about 65,000 samples. The main collectors are F. Conti (48,723), F. Bartolucci (12,539), D. Tinti (7193), A. Manzi (3254), D. Lakusic (3207) and D. Uzunov (2584). Many samples were collected together or with two or three collectors, so the sum of the specimens by a single collector exceeds the total number of samples. We must also remember some collectors such as G. Tondi, P. Minghetti and R. Soldati, whose collections, although conspicuous, have not yet been completely digitized. The taxonomic and chorological research concerning the endemic species of central Italy, focused on the Apennines, represents one of the most interesting peculiarities of the CRFA. Research work in this field can be expressed by the number of type specimens kept in APP (Table 1), many of which are used to describe new species discovered during the years of research activity of the CRFA. In APP herbarium, 146 type specimens are stored, among which 115 represent isotypes and 23 are holotypes; finally, two are paratypes and three are epitypes, while the remaining samples are one neotype, one isoneotype and one isoepitype. The type specimens belong to 60 taxa (species and subspecies).

**Table 1.** List of type specimens housed in APP.

| Taxon | Holotype | Isotype | Others |
|---|---|---|---|
| *Adonis fucensis* F.Conti & Bartolucci | 1 | | |
| *Allium ducissae* Bartolucci, Iocchi & F.Conti | 1 | 2 | |
| *Anthyllis apennina* F.Conti & Bartolucci | 1 | 6 | |
| *Anthyllis dalmatica* F.Conti & Stinca | 1 | 4 | |
| *Aquilegia magellensis* F.Conti & Soldano | 1 | | |
| *Centaurea arachnoidea* subsp. *montis-ferrati* Ricceri, Moraldo & F.Conti | 1 | 31 | |
| *Centaurea deusta* Ten. | | | Isoneotype |
| *Centaurea rupestris* L. | | | Epitype |
| *Centaurea valdemonensis* Domina, Di Grist., Barone | | 1 | |
| *Corydalis densiflora* subsp. *apennina* F.Conti, Bartolucci & Uzunov | 1 | 14 | |
| *Crepis magellensis* F.Conti & Uzunov | 1 | | |
| *Gelasia villosa* (Scop.) Cass. (≡ *Scorzonera villosa* Scop.) | | | Epitype, Isoepitype |
| *Genista pulchella* subsp. *aquilana* F.Conti & Manzi | 1 | 3 | |
| *Genista sericea* subsp. *pollinensis* F.Conti, Feoli Chiapella & Bernardo | 1 | 2 | |
| *Hieracium arpadianum* subsp. *pugnaculum* Gottschl. | | 1 | |
| *Hieracium boreoapenninum* Gottschl. | | | Paratype |
| *Hieracium cavallense* Gottschl. | | 1 | |
| *Hieracium contii* Gottschl. | | 1 | |
| *Hieracium galeroides* subsp. *aculeatisquamum* Gottschl. | | 1 | |
| *Hieracium glaucinum* subsp. *pseudobasalticum* Gottschl. | | 1 | |
| *Hieracium glaucinum* subsp. *tintiae* Gottschl. | | 1 | |
| *Hieracium hypochoeroides* subsp. *grandisaxense* Gottschl. | | 1 | |
| *Hieracium hypochoeroides* subsp. *pallidopsis* Gottschl. | | 1 | |
| *Hieracium hypochoeroides* subsp. *potamogetifolium* Gottschl. | | 1 | |
| *Hieracium latilepidotum* Gottschl. | | 1 | |
| *Hieracium lycopifolium* subsp. *ocreanum* Gottschl. | | 1 | |
| *Hieracium marsorum* Gottschl. | | 1 | |
| *Hieracium montis-porrarae* Gottschl. | | 1 | |
| *Hieracium neoplatyphyllum* subsp. *malacofloccosum* Gottschl. | | 1 | |
| *Hieracium neoplatyphyllum* subsp. *trimontanum* Gottschl. | | 1 | |
| *Hieracium nubitangens* Gottschl. | | | Paratype |
| *Hieracium pallidum* subsp. *lanudae* (Gottschl.) Gottschl., Raimondo & Di Grist. (≡ *Hieracium lanudae* Gottschl.) | | 1 | |
| *Hieracium permaculatum* Gottschl. | | 1 | |
| *Hieracium picenorum* Gottschl. | | 1 | |
| *Hieracium prenanthoides* subsp. *stupposifolium* Gottschl. | | 1 | |
| *Hieracium pseudogrovesianum* subsp. *amictum* Gottschl. | | 1 | |
| *Hieracium pseudogrovesianum* subsp. *circinans* Gottschl. | | 1 | |
| *Hieracium pseudogrovesianum* subsp. *leonense* Gottschl. | | 1 | |
| *Hieracium pseudopallidum* Gottschl. | | 1 | |
| *Hieracium schmidtii* subsp. *crinitisquamum* Gottschl. | | 1 | |
| *Hieracium simbruinicum* Gottschl. | | 1 | |
| *Hieracium venticaesum* Gottschl. | | 1 | |
| *Jurinea mollis* subsp. *mollis* f. *erectobracteata* F.Conti | 1 | | |
| *Lathyrus apenninus* F.Conti | 1 | | |
| *Mcneillia rosanoi* subsp. *moraldoi* (F.Conti) Del Guacchio & F.Conti (≡ *Minuartia moraldoi* F.Conti) | | 1 | |
| *Minuartia juniperina* subsp. *kosaninii* V.Stevanovic & Kamari | | 1 | |
| *Oxytropis ocrensis* F.Conti & Bartolucci | 1 | 2 | |
| *Pedicularis rostratospicata* subsp. *marsica* F.Conti & Bartolucci | 1 | | |
| *Pinguicula vallis-regiae* F.Conti & Peruzzi | 1 | | |
| *Pinguicula vulgaris* subsp. *anzalonei* Peruzzi & F.Conti | 1 | | |
| *Pinguicula vulgaris* subsp. *ernica* Peruzzi & F.Conti | 1 | | |
| *Pinguicula vulgaris* subsp. *vestina* F.Conti & Peruzzi | 1 | | |
| *Poa magellensis* F.Conti & Bartolucci | 1 | 1 | |

**Table 1.** *Cont.*

| Taxon | Holotype | Isotype | Others |
|---|---|---|---|
| *Ranunculus giordanoi* F.Conti & Bartolucci | 1 | 18 | |
| *Ranunculus multidens* Dunkel | | 1 | |
| *Reichardia albanica* F.Conti & D.Lakušic | 1 | 1 | |
| *Sedum aquilanum* L.Gallo & F.Conti | 1 | 1 | |
| *Senecio apenninus* Tausch | | | Neotype |
| *Siler montanum* subsp. *apuanum* F.Conti & Bartolucci | 1 | | |
| *Thymus spinulosus* Ten. | | | Paratype |

## 4. Discussion

As partially expressed by the taxonomic research and the geographical distribution of the collected specimens, one of the major interests of the CRFA is to evaluate the distribution of amphi-Adriatic species, their systematic relationships and the possible need to indicate additional features to better distinguish local populations. The APP herbarium is available to provide material for DNA extraction, and this is facilitated by the high proportion of recently collected samples, leaves or other specimen parts have already been sent to various research groups and, thanks to the collaboration developed with some of them, several systematic studies were carried out [21–24]. Another important aspect of interest is the synergy between herbarium activities, such as collection and management, and the preparation of the updates to checklists of the vascular flora of Italy. The extension of the geographical area explored by the CRFA operators, in addition to the abundant correspondences with national and international expert botanists, represent, on the one hand, a facilitation of the work of updating checklists at the national level. On the other hand, the constant work on the checklists of the Italian flora helps the same operators to outline an overall picture of the species and locations of greatest interest for floristic and taxonomic research in Italy. Ongoing and future works involving the CRFA, and the APP herbarium by consequence, will regard the flora of vascular plants of the Sirente Velino Natural Regional Park and the development of a regional database of the biodiversity and, in particular, of plants species, both vascular and not, in addition to the regular monitoring of more threatened and conservation relevant floristic taxa of the National Park of Gran Sasso and Monti della Laga. Other upcoming projects will concern in-depth studies, with an integrated taxonomic approach, of some critical taxa of phytogeographical interest in the context of oak woodlands of the Italian peninsula, as well as the publication of the herbarium collection data on international open-access platforms such as the "Global Biodiversity Information Facility" (GBIF) [25].

**Author Contributions:** Conceptualization and methodology, F.C., G.C. and F.B.; formal analysis, G.C.; data curation, F.C., G.C., J.D.V. and F.B.; writing—original draft preparation, F.C., G.C. and F.B.; writing—review and editing, F.C., G.C., J.D.V. and F.B.; supervision, F.C. and F.B. All authors have read and agreed to the published version of the manuscript.

**Funding:** This research received no external funding.

**Institutional Review Board Statement:** Not applicable.

**Data Availability Statement:** The original contributions presented in the study are included in the article, further inquiries can be directed to the corresponding author.

**Acknowledgments:** The authors wish to thank Cristina Blandino, Simone Cecchetti, Daniele Di Santo, Antea Gennari, Irene Londrillo, Joerg Meister, Paola Pavoni, Riccardo Pennesi, Elisa Proietti, Nicola Ranalli, Giovanni Santoni, Elisabetta Scassellati, Adriano Stinca, Sabrina Torcoletti, Donatella Vitale and Robert P. Wagensommer for contributing to the computerization of herbarium data and specimen management. We sincerely thank Daniela Tinti and Marco Iocchi, who developed the herbarium database.

**Conflicts of Interest:** The authors declare no conflicts of interest.

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
