# Peer review of "Herbarium Apenninicum (APP): An Archive of Vascular Plants from Central Italy"

_diversity, doi:10.3390/d16020099_

Round 1

Reviewer 1 Report

Comments and Suggestions for Authors

This article describes the features of the APP Herbarium. The main features of the APP Herbarium are described, which facilitates its use by researchers interested in it. Emphasis is placed on the geographical areas covered, on the presence of historical materials and on the type specimens preserved in the APP Herbarium.

Author Response

Thank you very much for the revision.

Reviewer 2 Report

Comments and Suggestions for Authors

It was a pleasure to read this manuscript that highlights floristic and taxonomic research and the current role of herbaria. The authors present the herbarium in the introduction, carefully describe the methodology and present the results based on the collections and treatment of the herbarium specimens.

In my opinion, the manuscript deserves to be published in Diversity after some minor revisions detailed below.

The introduction should emphasize the current importance of herbaria for studies of genomics, climate change, etc.

Lines 45-47. The descriptions mentioned by the authors refer to new taxa, according to the bibliography. Perhaps it should be highlighted in the introduction.

Lines 48-49. It would be nice to add the percentages to the absolute values.

Line 49 and others. When the authors cite the database, is it an open-access database? Does it include images of the vouchers or only the collection data? It would be advisable to provide a clearer explanation in the text.

Line 82. (PFI) should be added after “Portal of (of or to? ) the Flora of Italy”. The reader will come across this acronym later. The reference number 14 must be checked.

Lines 91-93. The sum of the percentages should be verified on these lines and also in Figure 2.

Figure 3. The Latin names should be in italics.

Subheading numbering must be revised. Three of them have 3.2.  

Figure 6 should have a discontinuous scale, the years 1975-2025 should be in the center.

Figure 7 is not necessary, it should be removed.

The discussion could be expanded with several topics. Does the herbarium offer any services such as material exchange, tissue bank or DNA extraction, among others, for external users? The APP herbarium preserves a significant collection of Hieracium species. Are there any forthcoming plans to undertake studies on this particularly challenging genus in collaboration with other genetics or genomics research groups, for instance?

Author Response

First of all thank you for the revision, you can see below the answers point by point to your comments.

- Introduction has been implemented with the suggested topics

- Lines 45-47: New taxa descriptions activities has been highlighted in the introduction

- Lines 48-49: We added the percentage value of the specimens entered in the database to the total of specimens preserved in APP herbarium

- Lines 49: Sadly the database is not yet open-access, one of the goal (as specified in the discussion) is to upload it on platforms such as GBIF.

- Line 82: PFI acronym has been added

- Lines 91-93: percentage values has been checked and fixed to reach the sum of 100%

- Figure 2: percentage values has been checked and fixed to reach the sum of 100%. 

- Figure 3: names of the genera has been set to italic

- Subheadings has been corrected

- Figure 6: we decided to change the graphic from histogram to barplot to group the data in the way we think most understandable for the reader. 
We aggregated the specimens collected before 1985 to leave more space to part of the graph showing more recent years.

- Figure 7: we agreed that it could be removed

- The discussion has been implemented with topics related to integrated taxonomy and DNA extraction activities that involves the APP herbarium and the CRFA in collaboration with other national and international research groups.

Reviewer 3 Report

Comments and Suggestions for Authors

The manuscript titled "Herbarium Apenninicum (APP): an Archive of Vascular Plants from Central Italy" falls within the scope of the Diversity journal. This work introduces the collection of vascular plants housed in the APP herbarium (located in Abruzzo, Italia), which has a substantial representation of vascular plant species from both Italy and other countries in the Mediterranean Basin. These studies are pivotal in highlighting the importance of botanical collections and their associated efforts and interest.

The manuscript's impact could be enhanced by incorporating references to cutting-edge research that has utilized the APP herbarium data. While optional, it is worth considering the removal of Figure 7, with the possibility of noting the number of specimens per collector within the text as a sufficient alternative. Additionally, it is highly recommended that the authors publish future herbarium collection data on open-access international platforms, such as GBIF (www.gbif.org).

Author Response

The discussion has been implemented including references to studies that has utilized APP data as you suggested.
We agreed with the removal of the Figure 7 and we accepted your advice to indicate the data directly in the text.
Finally we thank you for for having brought out another future goal of the herbarium that has been often discussed, namely its inclusion in GBIF platform.

Reviewer 4 Report

Comments and Suggestions for Authors

The manuscript for the communication type is recommended and may be published. I missed the conservation theme that could be explored with the information contained in the herbarium.

Little consideration was made in the text.

Author Response

- Line 54: yes, many specimens were collected as part of conservation projects. We added it in the text

- Lines 59-60: we were referring to national and international floras and literature specialized on specific plant groups (families, genera, etc..). Nevertheless, as also floras can be defined as specialized literature, we accepted your correction.

- Line 90: we prefer to leave "species and subspecies" because taxa would also include other taxonomic categories (even more detailed) and according to several authors it is too generic and unclear

- Discussion: the APP herbarium and, even more, the CRFA are regularly involved in projects dealing with conservation of biodiversity. These are part, however, of a type of activities that are ordinary for the research center. One of it is the monitoring of threatened and rare palnt species of the National Park of Gran Sasso and Monti della Laga, but also the preparation of the floras of other protected areas can be seen as a conservation goal.
Even the taxonomical studies concerning amphiadriatic species can have outcomes that foster biodiversity conservation by, for example, the discovery of new endemic taxa (for example subspecies of amphiadriatic species).